# Advancing Prompt Learning through an External Layer

Fangming Cui
Shanghai Jiaotong University
Shanghai, China
Defense Innovation Institute
Beijing, China
cuifangming@sjtu.edu.cn

Xun Yang
MoE Key Laboratory of
Brain-inspired Intelligent Perception
and Cognition, University of Science
and Technology of China
Hefei, China
xyang21@ustc.edu.cn

Chao Wu
Zhejiang University
Hangzhou, China
chao.wu@zju.edu.cn

Liang Xiao*
Defense Innovation Institute
Beijing, China
Intelligent Game and Decision
Laboratory
Beijing, China
xiaoliang@nudt.edu.cn

Xinmei Tian
MoE Key Laboratory of
Brain-inspired Intelligent Perception
and Cognition, University of Science
and Technology of China
Hefei, China
xinmei@ustc.edu.cn

## ABSTRACT

Prompt learning represents a promising method for adapting pre-trained vision-language models (VLMs) to various downstream tasks by learning a set of text embeddings. One challenge inherent to these methods is the poor generalization performance due to the invalidity of the learned text embeddings for unseen tasks. A straightforward approach to bridge this gap is to freeze the text embeddings in prompts, which results in a lack of capacity to adapt VLMs for downstream tasks. To address this dilemma, we propose a paradigm called *EnPrompt* with a novel *External Layer* (EnLa). Specifically, we propose a textual external layer and learnable visual embeddings for adapting VLMs to downstream tasks. The learnable external layer is built upon valid embeddings of pre-trained CLIP. This design considers the balance of learning capabilities between the two branches. To align the textual and visual features, we propose a novel two-pronged approach: i) we introduce the optimal transport as the discrepancy metric to align the vision and text modalities, and ii) we introduce a novel strengthening feature to enhance the interaction between these two modalities. **Four** representative experiments (i.e., base-to-novel generalization, few-shot learning, cross-dataset generalization, domain shifts generalization) across **15** datasets demonstrate that our method outperforms the existing prompt learning method.

## CCS CONCEPTS

• **Computing methodologies → Artificial intelligence**; **Machine learning**.

*Corresponding author.

## KEYWORDS

Vision-language Model, Prompt Learning, Optimal Transport

**ACM Reference Format:**
Fangming Cui, Xun Yang, Chao Wu, Liang Xiao, and Xinmei Tian. 2024. Advancing Prompt Learning through an External Layer. In *Proceedings of Proceedings of the 32nd ACM International Conference on Multimedia (MM '24)* ACM, New York, NY, USA, 10 pages. https://doi.org/10.1145/3664647.3680953

## 1 INTRODUCTION

Vision-language models (VLMs), such as CLIP [57] and ALIGN [33], have demonstrated remarkable generalization performance for various downstream tasks. VLMs are typically trained to align textual and visual modalities using large-scale datasets, which allows them to encode open-vocabulary [20, 21] concepts in a shared embedding space. Thanks to the modality matching ability, CLIP has achieved remarkable success across various downstream tasks [27, 28, 48–50, 80], such as action recognition [66], image generation [60], and image segmentation [46, 79]. One of the attractive features of CLIP is the ability to perform zero-shot inference, where some pre-defined text inputs (a.k.a. prompts) are used to generate classification weights for predicting image features during inference.

Seminal explorations involve hand-crafted text prompts, e.g., 'a photo of a [class]' as the prompt for text encoder. Advanced works propose to introduce a set of learnable parameters to adapt VLMs to downstream tasks. For instance, CoOp [82] keeps the weights of CLIP frozen while learning the text embeddings of prompts for efficient task-specific adaptation. These prompt-learning approaches achieve significant performance improvements over manually tuned prompts on various typical scenarios [7, 11, 17]. However, the text embeddings learned typically produce worse performance compared with hand-crafted prompts [57] for unseen tasks. Specifically, applying these learned text prompts to unseen classes leads to degenerated model generalization performance. Recent works understand and mitigate this problem through the overfitting view. To overcome the challenge, an image-conditional prompt (CoCoOp) learning approach [81] is proposed to promote the generalization performance in unseen classes, which is higher

than CoOp. Unfortunately, CoCoOp exhibits lower generalization performance than hand-crafted zero-shot CLIP when applied to novel classes, as depicted in Table 4. This could be attributed to the learned embeddings are usually invalid in novel classes.

To address this dilemma, we propose a paradigm called EnPrompt with a novel External Layer (EnLa). Specifically, we propose textual external layer and learnable visual embeddings for adapting VLMs to downstream tasks through freezing the text embeddings. To align the textual and visual features, we introduce the optimal transport as a discrepancy metric that measures the difference between the visual and textual features, where the optimal transport can calculate the distance between two distributions under the form of multiple sampling. Further, we connect the text and image encoders through a strengthening feature instead of ordinary coupling for the dimension transformation of learnable hidden vectors. As shown in Figure 1, EnPrompt outperforms the existing state-of-the-art method in base-to-novel generalization task. Our main contributions can be summarized as follows:

- We address the performance-degeneration issue by introducing a novel External Layer (EnLa) and freezing the text embeddings.
- We align the visual and textual features of the two modalities by introducing a novel two-pronged approach. Specifically, we introduce the optimal transport as the discrepancy metric to measure and mitigate the difference between the visual and textual features. Meanwhile, we introduce a novel interaction strategy between modalities to strengthen the modality fusion.
- **Four** highly representative experiments demonstrate that our method can consistently outperform the existing methods across **15** datasets, achieving state-of-the-art performance.

## 2 METHODOLOGY

### 2.1 Preliminaries

We provide a brief introduction to vision-language pre-training, with a specific focus on CLIP, which is a zero-shot learning approach without fine-tuning. We build our approach based on CLIP. CLIP has a text encoder $\mathcal{F}_t(\cdot)$ and an image encoder $\mathcal{F}_v(\cdot)$, which separately map a textual input (a.k.a. prompt) $\mathbf{p}$ and a visual input, i.e., an image, $\mathbf{x}$ into a shared feature space through many transformer blocks. The outputs of two encoders are denoted as $t = \mathcal{F}_t(\mathbf{p})$ and $v = \mathcal{F}_v(\mathbf{x})$. The image encoder aims to transform the input images into feature embeddings, while the text encoder generates representations for word embedding sequences.

During the pre-training phase of CLIP, these two encoders are simultaneously trained on large-scale datasets of text-image pairs, where a contrastive loss is employed to maximize the cosine similarity of text-image pairs and minimize the cosine similarity between unmatched pairs in the feature space. The final prediction probability of alignment is computed by the matching score as follows:

$$p(y \mid \boldsymbol{x}, \mathcal{P}) = \frac{\exp\left\{sim\left(\mathcal{F}_t\left(\mathbf{p}_y\right), \mathcal{F}_v(\mathbf{x})\right) / \tau\right\}}{\sum_{\mathbf{p}_i \in \mathcal{P}} \exp\left\{sim\left(\mathcal{F}_t\left(\mathbf{p}_i\right), \mathcal{F}_v(\mathbf{x})\right) / \tau\right\}}, \quad (1)$$

where $y \in \mathcal{Y}$ is the label of $\mathbf{x} \in \mathcal{X}$, $\mathcal{P} = \{\mathbf{p}_i\}_{i=1}^{C}$ denotes the set of $C$ pre-defined prompts, $sim(\cdot, \cdot)$ stands for cosine similarity between

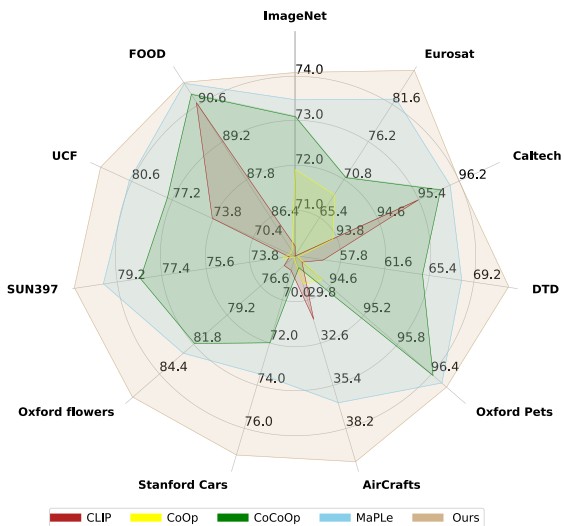

**Figure 1: Performance comparison on base-to-novel generalization. EnPrompt (Ours) outperforms previous state-of-the-art methods on 11 datasets.**

two vectors, and $\tau > 0$ represents a temperature parameter. Here, the classifier consists of $C$ textual features derived from pre-defined prompts $\mathcal{P} = \{\mathbf{p}_i\}_{i=1}^{C}$, where the prompt $\mathbf{p}_i$ for the $i$-th class may have the form of 'a photo of a dog'.

Advanced works take a further step to investigate the possibility of aligning images and prompts. The insight of these works is that the prompts tuning for given images could be superior to hand-crafted prompts. Specifically, the class name is retained as prior knowledge to ensure the learned prompts can form a classifier, while the word (a.k.a. context) embeddings of prompts are modeled as learnable parameters. Here, the learnable words in the above prompt are typically initialized using 'a photo of a'. The embeddings optimization approach can be formalized as follows,

$$\min_{\mathbf{w}} \ell(\mathbf{w}) = -\log p(y \mid \mathbf{x}, \mathcal{P}(\mathbf{w})), \quad (2)$$

where $\mathbf{w}$ stands for learnable embeddings used to model the context in prompts. The prompt $\mathbf{p}_i \in \mathcal{P}(\mathbf{w})$ learned by the context optimization approach may have the form of,

$$\mathbf{p}_i := [\mathbf{w}] \, \textcircled{c} \, [ClassName_i], \quad (3)$$

where $\textcircled{c}$ is a concatenating operation. Here, some learned text embeddings methods follow CoOp [82] to set the embeddings $\mathbf{w}$ shared across classes. In the inference phase, the prompts with the learned embeddings can produce textual features for classification.

### 2.2 EnLa with Frozen Text Embeddings

As depicted in Table 4, the existing methods CoOp [82] and Co-CoOp [81] work well on the base classes observed during training and beat the hand-crafted prompts method employed by CLIP [57] by a significant margin. However, these methods typically perform worse than hand-crafted prompts for novel classes (unseen tasks). This may be attributed to the non-adaptive of the learned text embeddings on unseen tasks. Learning task-specific text embeddings

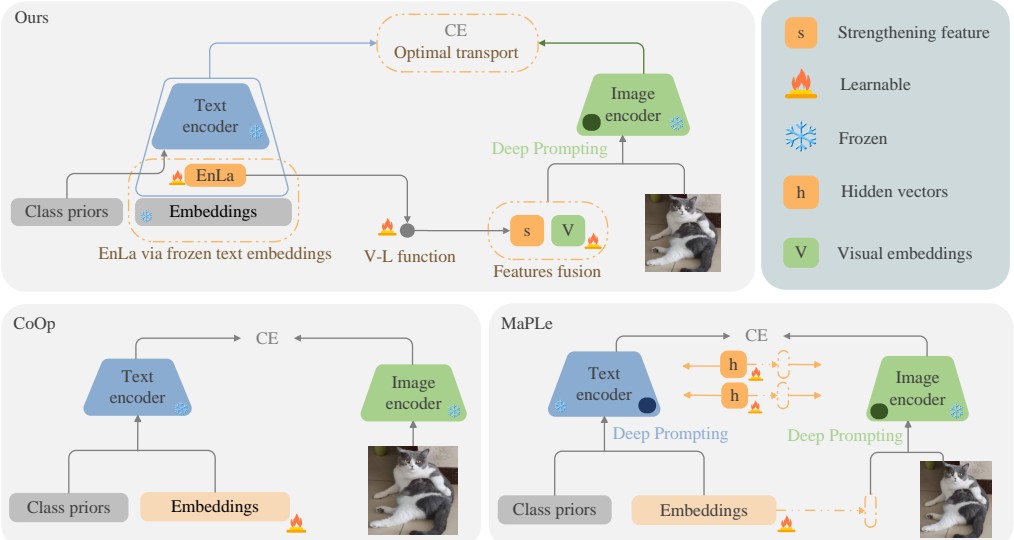

**Figure 2: EnPrompt (Ours) aims to connect the image and text encoders by a novel design, which introduces the strengthening feature of the EnLa to fuse the learnable visual embeddings of the first layer of image encoder, while frozen text embeddings. Both CoOp and MaPLe employ designs to learn text embeddings. In our method, the fusion embeddings are further introduced in transformer of the image encoder for deep prompting.**

can result in the loss of the essential general textual knowledge having a strong generalization ability.

**Frozen Text Embeddings.** Built upon the observation, we propose to freeze the text embeddings of prompts for unseen tasks. Specifically, the text embeddings are set to be frozen and non-learnable, which aims to release the potential generalization ability of CLIP and ensure pre-trained ability for unseen tasks. Here, the input words in the prompt are typically initialized using 'a photo of a', which can be vectorized into embeddings. In our upcoming ablation experiments, we plan to incorporate other manual template inputs for validation. However, freezing text embeddings of prompts will cause limited model capacity, which results in a lack of capacity to adapt VLMs for downstream tasks.

**External Layer (EnLa).** To address this dilemma, we propose to introduce an External Layer (EnLa) of the text branch and learnable visual embeddings of the visual branch to learn with VLMs for adapting downstream tasks. This design considers the balance of learning capabilities between the two branches. As depicted in Figure 2, EnLa can be considered as an extension of the text encoder of VLMs, which is an auxiliary layer introduced to the architecture. Introducing the EnLa will make VLMs learnable while keeping the text embeddings frozen. In contrast to CLIP [57], our method is being explored in the field of few-shot learning.

Let $\mathcal{E}_{\theta}(\cdot)$ denote the EnLa parameterized by $\theta$, which is updated for adapting VLMs to downstream tasks. Moreover, let $\mathcal{P}$ denote the frozen text embeddings. The $\mathcal{E}_{\theta}(\cdot)$ is also realized as an incomplete embedding transformer, showing the same spirit with neural networks. Specifically, let $e$ denote the output of the EnLa,

$$e = \mathcal{E}_{\theta}(\mathcal{P}), \tag{4}$$

where $e$ is further transferred as input to two directions: text encoder $\mathcal{F}_t(\cdot)$ and vision-language interaction, particularly towards the image encoder $\mathcal{F}_v(\cdot)$.

**EnLa Design Analysis.** We compare the performance of a single layer with a two-layer, referring to the bottleneck structure (Linear-ReLU-Linear), on 11 datasets with 16 shots for base-to-novel tasks. HM refers to the harmonic mean. The results in Table 1 indicate that the single 512x512 structure (row-5) provides better performance. It achieves the highest HM of 80.64%, which surpasses all reduction factors of the hidden layers that have 32x, 16x, 8x, and 4x. Thus, we used a single layer of EnLa in all of our experiments.

**Table 1: Ablation study on the EnLa designs.**

| Layer Design | Reduction factor | Base Acc. | Novel Acc. | HM |
|---|---|---|---|---|
| 1: (512 x 16) (16 x 512) | 32x | 83.9 | 73.40 | 78.3 |
| 2: (512 x 32) (32 x 512) | 16x | 84.6 | 76.8 | 80.51 |
| 3: (512 x 64) (64 x 512) | 8x | 84.45 | 76.70 | 80.41 |
| 4: (512 x 128) (128 x 512) | 4x | 84.40 | **77.0** | 80.54 |
| 5: (512 x 512) | / | **84.71** | 76.90 | **80.64** |

## 2.3 Alignment with Optimal Transport

To align two modalities for generalization, we introduce the optimal transport as a discrepancy metric and formulate the feature sets as discrete probability distributions for generalization performance. In contrast to conventional distance metrics such as Euclidean distance, optimal transport [64] learns an adaptive transport plan to calculate the cross-modality distance, facilitating fine-grained matching across the two modalities. It enables us to align visual features in a more precise and adaptive manner, capturing subtle nuances and enhancing the matching accuracy between modalities.

 Fangming Cui et al.

**Optimal Transport.** The Optimal Transport (OT) distance is a commonly employed metric for comparing distributions. In the context of our framework, we specifically concentrate on the discrete situation as it aligns more closely with our approach. In this scenario, we consider two sets of points or features.

Given two sets that contain $N$ and $M$ points respectively, the discrete distributions can be formulated as follows:

$$\mathbf{Z}_\theta = \sum_{n=1}^{N} \theta_n \delta_{e_n} \quad \text{and} \quad \mathbf{Q}_\beta = \sum_{m=1}^{M} \beta_m \delta_{l_m} \quad (5)$$

where $\theta \in \Delta^N$ and $\beta \in \Delta^M$ are discrete probability vectors that sum to 1 , and $\delta_e$ refers to a point mass located at point $e$ in the embedding space. The OT distance between $\mathbf{Z}_\theta$ and $\mathbf{Q}_\beta$ is defined as:

$$d_{\text{OT}}(\theta, \beta) := \min_{\mathbf{T}} < \mathbf{T}, \mathbf{C} >, \quad (6)$$

$$\mathbf{T}\mathbf{1}_M = \theta, \quad \mathbf{T}^\top \mathbf{1}_N = \beta, \quad (7)$$

where $< \cdot, \cdot >$ denotes the Frobenius dot-product and $\mathbf{1}_N$ is the $N$ dimensional vector of ones. $\mathbf{C} \in \mathbb{R}_{>0}^{N \times M}$ is the cost matrix of the transport, and $C_{nm}$ denotes the transport cost between points $e_n$ and $l_m$, such as the cosine distance $C_{nm} = 1 - \text{cosine}(e_n, l_m)$ .$\mathbf{T} \in \mathbb{R}_{>0}^{N \times M}$ denotes the transport plan to be learned. OT distance is then minimized over all the joint probabilities of $N \times M$ space with two marginal constraints. As computing the above OT distance has the cubic time complexity, we apply the Sinkhorn distance [13] that regularizes with an entropic constraint:

$$d_{\text{OT},\lambda}(\theta, \beta) = d_{\text{OT}}(\theta, \beta) - \lambda h(\mathbf{T}), \quad (8)$$

$$\text{with} \quad \mathbf{T}\mathbf{1}_M = \theta, \quad \mathbf{T}^\top \mathbf{1}_N = \beta \quad (9)$$

where $h(\mathbf{T})$ is the entropy of transport plan $\mathbf{T}$ and $\lambda \geq 0$ is a hyper-parameter. It can be optimized within a few iterations by the Sinkhorn algorithm with the Lagrange multiplier of the entropy constraint.

**Modalities Alignment with Optimal Transport.** The transport plan is efficiently computed through a limited number of matrix multiplications as a forward module. These matrix multiplications are crucial for determining the gradients that are then preserved for back-propagation.

Specifically, let $v$ denote the visual feature and $t$ denote the text feature. The output of the image encoder is a tensor with the shape, where $H$ and $W$ are the height and width. Therefore, we can obtain $M = H \times W$ local visual features. Further, let $V = \left\{ v_m |_{m=1}^M \right\}$ denote the local visual features as the fixed set, let $t_k$ denote the text feature as the fixed set for class $k$. Our method learns the transport plan $\mathbf{T}$ by minimizing the following OT distance to push $t_k$ to $V$ for fine-grained alignment:

$$d_{\text{OT}}(k) = d_{\text{OT}} \left( \theta, \beta \mid 1 - V^\top t_k \right), \quad (10)$$

where $C = 1 - V^\top t_k$ denotes that we use the cosine distance between $V$ and $t_k$ as the cost matrix. Then, we can obtain the solution of transport plan $\mathbf{T}^*$ and the final OT distance $d_{\text{OT}}(k)$. Given the OT distance between $t_k$ and $V$, and image $x$, we reformulate the final prediction probability of V-L alignment as follows:

$$p_{\text{OT}}(y = k \mid x) = \frac{\exp\left( (1 - d_{\text{OT}}(k)) / \tau \right)}{\sum_{k'=1}^{K} \exp\left( (1 - d_{\text{OT}}(k')) / \tau \right)}. \quad (11)$$

In our method, we first fix the visual and textual features to optimize the optimal transport problem to calculate the cross-modality distance, obtaining the transport plan $\mathbf{T}^*$. Further, we back-propagate the gradient with cross-entropy loss to learn the learnable parameters of our method by fixing $\mathbf{T}^*$. This process is more robust to variations in visual domain shift tasks and tolerant to generalization.

## 2.4 Alignment with Strengthening Feature

In order to align two modalities for unseen tasks, we propose to connect the visual and textual modalities through a novel strengthening feature with a vision-language (V-L) function instead of coupling learnable hidden vectors for V-L alignment.

We introduce learnable visual embeddings for enhancing learnality of VLMs for downstream tasks. In contrast to VPT [34], our method is being explored in the multi-modal prompt designs. Specifically, let $\mathcal{T}_\epsilon(\cdot)$ denote the V-L function parameterized by $\epsilon$ to transfer features from text branch to visual branch. The $\mathcal{T}_\epsilon(\cdot)$ is a function net with dimensionality of [512, 768] for dimension alignment of the visual branch and text branch. The V-L function will modify visual embeddings using the features generated by the EnLa, the input feature of the V-L function is the output produced by the EnLa. Let $e$ denote the output of EnLa, Further, we append learnable $L$ visual input embeddings $\mathcal{P}_v$, which is the vector with $[L, 768]$, given as follows,

$$\mathcal{P}_v = \left\{ p_v^1, p_v^2, \cdots, p_v^L \right\}. \quad (12)$$

To improve generalization capabilities, we are conducting an evaluation to assess the performance of different connection positions of the image encoder, as depicted in Table 2.

**Table 2: Different positions of strengthening feature in base-to-novel generalization. Text embeddings frozen based on EnLa is the default setting. Results are averaged over 11 datasets. HM refers to the harmon.**

| Position of Strengthening Feature | Base Acc. | Novel Acc. | HM |
|---|---|---|---|
| 1: Deep layer of image encoder | 83.85 | 76.31 | 79.90 |
| 2: Input layer of image encoder | **84.71** | **76.90** | **80.64** |

The hidden layer of the image encoder (row-1) combines the benefits of prompting in both branches by enforcing the image encoder representation ability through text knowledge transfer to the deep layer of the image encoder without visual embeddings initialization layer, and the input layer of the image encoder (row-2) is based on the connection of the visual embeddings initialization. The position (row-2) is more effective than the position (row-1) in unseen tasks. It can be attributed to that position (row-1) lacks visual embedding initialization, leading to the acquisition of a limited ability to adapt the image encoder with the unbalance of V-L alignment. As a consequence, in our subsequent comprehensive experiments, we adopted the connection position with the input layer of the image encoder (row-2) in our approach. To this end, let $p_v(e)$ denote the fusion embeddings, as follows,

$$p_v(e) = p_v + \mathcal{T}_\epsilon(e). \quad (13)$$

The fusion embeddings are further introduced in the transformer of the image encoder for deep prompting.

In contrast to EnPrompt (Ours), CoCoOp transfers the output features of the image encoder to the text embeddings, which is more inference time consumption. MaPLe initializes the multi-learnable hidden vectors for V-L alignment and couples the text embeddings to the visual encoder. The coupling function maps these initialized hidden vectors of deep layer and text embeddings with the image encoder and text encoder for more parameters. However, our approach introduces the learnable visual initialization embeddings of the image encoder with the strengthening feature fusion. However, MaPLe learns to visual branch with the coupled visual embeddings without the visual embeddings initialization step, which limits the ability of the visual branch to adapt downstream data distributions.

## 3 EXPERIMENTS

### 3.1 Benchmark Setting

We compare the performance with CLIP [57], CoOp [82], Clip-Adapter [26], CoCoOp [81], PLOT [8], and MaPLe [35]. Please note that CLIP was originally a zero-shot method. However, in this context, we linearly process it using a few-shot method. The Clip-Adapter [26] is a frozen prompt method with the adapter component behind the encoder, which is different from EnPrompt. We compared it with PLOT [8], which is an optimal transport method. PLOT optimizes four sentences simultaneously ("a photo of a dog", "a picture of a dog", "a drawing of a dog", "a good drawing of a dog"). The output text feature of EnPrompt is global feature, the input sentence is one text prompt such as "a photo of a", and the text prompt is non-learnable.

**Few-shot Learning.** We employ this setup to evaluate En-Prompt in conditions of highly restricted supervision. We evaluate the model on 11 datasets by conducting tests with varying K-shots per class, where K takes the values of 1, 2, 4, 8, and 16.

**Base-to-Novel Generalization.** We evaluate the ability to generalize following the setting where the datasets are split into base and novel classes. The model is trained only on the base classes in 16 shots setting and evaluated on base and novel classes.

**Cross-dataset Evaluation.** To validate the potential of our approach in cross-dataset transfer, we evaluate our ImageNet-trained model in 16 shots directly on other 10 datasets. This experiment aims to verify whether our model can successfully complete the unseen generalization.

**Domain Generalization.** We evaluate the robustness of our approach to domain shift datasets. Similar to cross-dataset evaluation, we train our model using the ImageNet in 16 shots and evaluate its performance directly on 4 different variants of the ImageNet.

**Datasets.** For base to novel class generalization, few-shot settings, and cross-dataset evaluation, we use 11 image recognition datasets. The datasets cover multiple recognition tasks including ImageNet [14] and Caltech101 [25] which consists of generic objects, OxfordPets [56], StanfordCars [36], Flowers102 [54], Food101 [4], and FGVCAircraft [52] for fine-grained classification, SUN397 [67] for scene recognition, UCF101 [62] for action recognition, DTD [12]

for texture classification, and EuroSAT [30] which consists of satellite images. For domain generalization benchmark, we use ImageNetA [32], ImageNet-R [31], ImageNet-Sketch [65] and ImageNetV2 [58] as domain shift datasets.

**Implementation Details.** We use a ViT-B/16-based CLIP model, and report results averaged over 3 runs. We set the learnable visual embedding length to 4. We use deep prompting with an image encoder. Training for 50 epochs for few-shot setting, 20 epochs for domain generalization setting and cross-dataset evaluation setting. In the base-to-novel setting, we use 30 epochs for most datasets. We use an SGD optimizer with a learning rate of 0.0025. All experiments are run on a single Nvidia A6000 GPU. For domain generalization and cross-dataset evaluation, we train the ImageNet source model on all classes with $K = 16$ shots in the first 3 transformer layers of the image encoder. For the few-shot learning and base-to-novel tasks, we set visual embeddings learning depth to 9. We initialize the text embeddings using the hand-crafted words of 'a photo of a'.

### 3.2 Effect of Components

In Table 3, we disentangle the components and show the individual contributions to the base-to-novel generalization task. Results are averaged over 11 datasets with 16-shot. HM refers to harmonic mean. Integrating EnLa (row-2) outperforms baseline methods (row-1) on novel classes while maintaining base class gains. By enforcing visual embeddings (row-3), HM performance significantly increases due to the ability to adapt the image encoder for better V-L alignment. Integrating optimal transport (row-4) outperforms component (row-3) with an improvement of 1.08% in HM. This suggests that the regularization of OT is more effective than the traditional metric function in calculating the cross-modality distance. Finally, combined with strengthening features and connecting to overcome the mismatch between the text and visual branch (row-5), our method achieves improvements on both base and novel classes.

**Table 3: Effect of our components in base-to-novel generalization. Results are averaged over 11 datasets.**

| Components | Base Acc. | Novel Acc. | HM |
|---|---|---|---|
| 1: Frozen text embeddings | 69.34 | 74.22 | 71.70 |
| 2: + EnLa | 80.24 | 74.43 | 77.25 |
| 3: + Visual embedding learning | 82.25 | 75.94 | 78.98 |
| 4: + Optimal transport | 84.50 | 76.05 | 80.06 |
| 5: + Strengthening feature | **84.71** | **76.90** | **80.64** |

### 3.3 Few-shot Learning Experiments

EnPrompt (Ours) consistently provides improvements on all few-shot settings compared to existing approaches. We note that it is effective for EnPrompt to enhance the few-shot image recognition task through our novel designs. Furthermore, we note that EnPrompt achieves relatively larger gains in minimal data cases, such as for $K = 2$ for almost all datasets. This finding suggests that EnPrompt achieves better alignment between the visual and language branches in seen class tasks, even with limited training resources. Moreover, EnPrompt demonstrates significant improvements compared to linear probe CLIP.

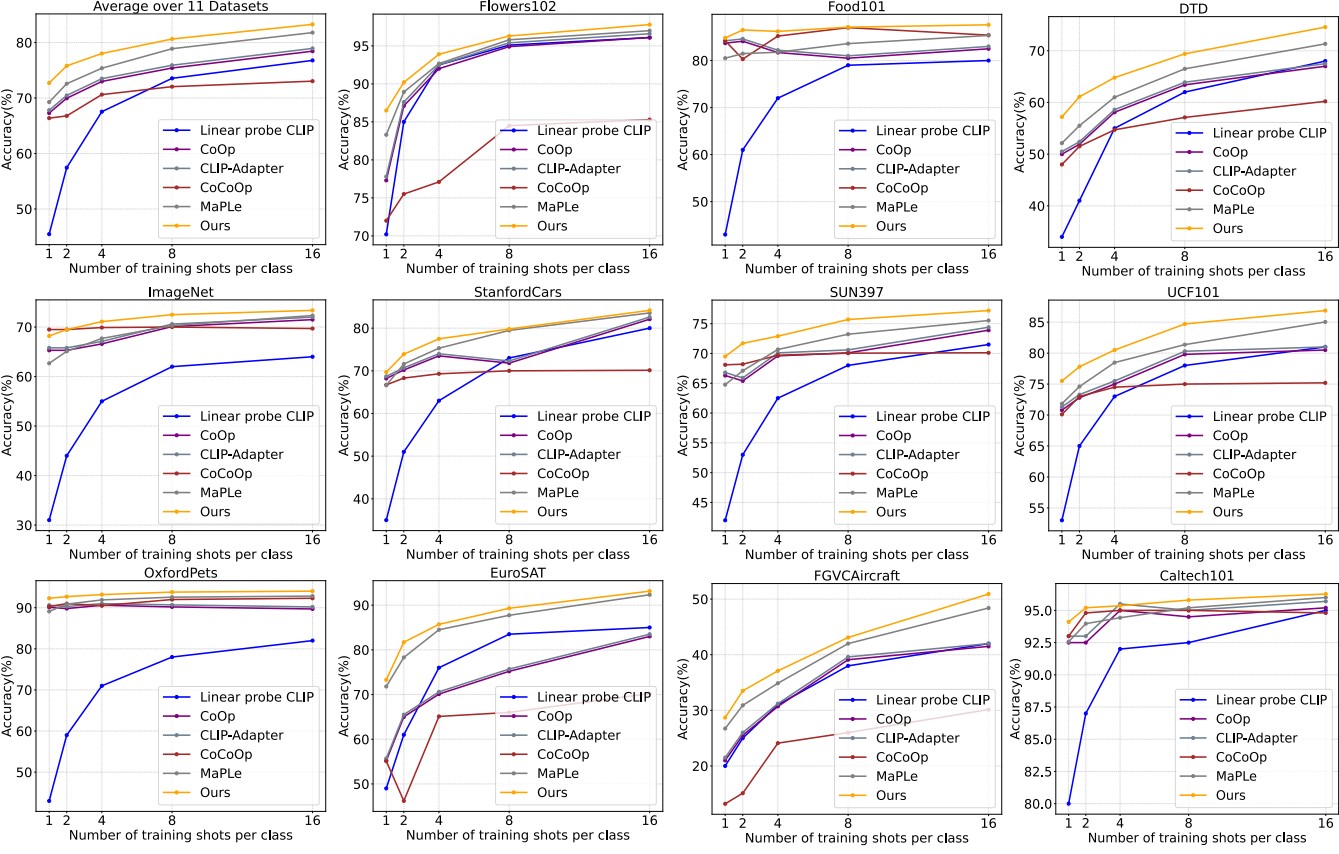

**Figure 3: Few-shot Learning Experiments. All methods are trained on the ViT-B/16 CLIP backbone. EnPrompt (Ours) demonstrates consistent improvements over existing methods, specifically for minimal training data such as K=2. On average, EnPrompt provides the highest performance gains for all shots (K = 1,2,4,8,16).**

## 3.4 Base-to-Novel Generalization Experiments

In Table 4, EnPrompt (Ours) demonstrates significant improvements on all 11 datasets. In general, existing approaches CoOp and CoCoOp demonstrate better performance than zero-shot CLIP on base classes. However, when it comes to novel classes, these approaches tend to exhibit normal performance. However, EnPrompt significantly enhances the performance on base classes while also improving the accuracy of zero-shot CLIP on novel classes by 2.69%. Overall, EnPrompt provides the best-averaged results of 84.71%, 76.90%, and 80.64% on the base classes, novel classes, and harmonic mean, respectively.

## 3.5 Cross Dataset Evaluation

To verify the cross-dataset generalization ability, we train EnPrompt (Ours) on the ImageNet dataset with 1,000 classes, and test it on the remaining 10 datasets. As shown in Table 5, EnPrompt shows competitive performance and achieves better generalization in 8/10 over the CoCoOp. Compared with MaPLe, EnPrompt achieves better performance. This indicates that EnPrompt favors better generalization for a wide range of datasets.

## 3.6 Domain Generalization Experiments

In Table 6, EnPrompt (Ours) consistently outperforms all existing methods on target datasets, with an overall highest average accuracy of 60.74%. EnPrompt achieves average gains of +1.46% over the CoOp. Compared with MaPLe, EnPrompt shows improved performance in all ImageNet variants datasets. This suggests that EnPrompt is focused on improving generalization capability for domain shifts. Furthermore, EnPrompt provides the highest accuracy of 51.45% on ImageNetA [32].

## 4 ABLATIVE ANALYSIS

### 4.1 Comparison of Different Templates

We conducted an experiment using various templates as inputs to the model. Remarkably, we observed that the HM values generated by different templates were very close to each other (see Table 7). The prompt templates can be relatively diversified because of the relatively stronger learning ability of EnLa on unseen tasks, due to the black box [76] nature of EnLa models.

**Table 4: Base-to-novel generalization experiments. EnPrompt (Ours) demonstrates strong generalization results over existing methods on 11 recognition datasets. Here, the CLIP refers to the linear probe CLIP.**

| Dataset | | CLIP | CoOp | CLIP-Adapter | CoCoOp | PLOT | MaPLe | Ours | Δ |
|---|---|---|---|---|---|---|---|---|---|
| | Base | 69.34 | 82.69 | 82.91 | 80.47 | 81.3 | 82.28 | **84.71** | +2.4 |
| Average | Novel | 74.22 | 63.22 | 63.98 | 71.69 | 72.2 | 75.14 | **76.90** | +1.8 |
| | HM | 71.70 | 71.66 | 72.23 | 75.83 | 76.48 | 78.55 | **80.64** | +2.1 |
| | Base | 72.43 | 76.47 | 76.88 | 75.98 | 75.33 | 76.66 | **77.70** | +1.1 |
| ImageNet | Novel | 68.14 | 67.88 | 68.1 | 70.43 | 70.48 | 70.54 | **70.65** | +0.1 |
| | HM | 70.22 | 71.92 | 72.23 | 73.10 | 72.83 | 73.47 | **74.07** | +0.6 |
| | Base | 96.84 | 98.00 | 98.1 | 97.96 | 97.86 | 97.74 | **98.40** | +0.7 |
| Caltech101 | Novel | 94.00 | 89.81 | 90.00 | 93.81 | 93.99 | **94.36** | 94.07 | -0.3 |
| | HM | 95.40 | 93.73 | 93.89 | 95.84 | 95.92 | 96.02 | **96.2** | +0.2 |
| | Base | 91.17 | 93.67 | 93.88 | 95.20 | 95.7 | 95.43 | **95.67** | +0.2 |
| OxfordPets | Novel | 97.26 | 95.29 | 95.55 | 97.69 | 98.1 | **97.76** | 97.63 | -0.1 |
| | HM | 94.12 | 94.47 | 94.74 | 96.43 | 96.80 | 96.58 | **96.67** | +0.1 |
| | Base | 63.37 | 78.12 | 78.35 | 70.49 | 71.5 | 72.94 | **78.70** | +5.8 |
| StanfordCars | Novel | 74.89 | 60.40 | 60.55 | 73.59 | 73.77 | 74.00 | **75.67** | +1.6 |
| | HM | 68.65 | 68.13 | 68.33 | 72.01 | 72.62 | 73.47 | **77.22** | +3.8 |
| | Base | 72.08 | 97.60 | 97.61 | 94.87 | 95.1 | 95.92 | **98.47** | +2.5 |
| Flowers102 | Novel | 77.80 | 59.67 | 59.98 | 71.75 | 72.2 | 72.46 | **77.00** | +4.4 |
| | HM | 74.83 | 74.06 | 74.32 | 81.71 | 82.10 | 82.56 | **86.43** | +4.0 |
| | Base | 90.10 | 88.33 | 88.55 | 90.70 | 90.98 | 90.98 | **91.00** | +0.3 |
| Food101 | Novel | 91.22 | 82.26 | 82.35 | 91.29 | 91.54 | **92.05** | 91.80 | -0.9 |
| | HM | 90.66 | 85.19 | 85.36 | 90.99 | 91.28 | 91.38 | **91.41** | +0.1 |
| | Base | 27.19 | 40.44 | 40.66 | 33.41 | 35.6 | 37.44 | **43.27** | +5.8 |
| FGVCAircraft | Novel | 36.29 | 22.30 | 23.1 | 23.71 | 28.5 | 35.61 | **37.77** | +2.0 |
| | HM | 31.09 | 28.75 | 29.46 | 27.74 | 31.66 | 36.50 | **40.34** | +3.7 |
| | Base | 69.36 | 80.60 | 80.85 | 79.74 | 79.96 | 80.82 | **82.77** | +2.0 |
| SUN397 | Novel | 75.35 | 65.89 | 65.91 | 76.86 | 77.33 | 78.70 | **79.07** | +0.3 |
| | HM | 72.23 | 72.51 | 72.62 | 78.27 | 78.64 | 79.75 | **80.91** | +1.2 |
| | Base | 53.24 | 79.44 | 80.56 | 77.01 | 78.9 | 80.36 | **83.87** | +3.5 |
| DTD | Novel | 59.90 | 41.18 | 45.30 | 56.00 | 57.9 | 59.18 | **63.67** | +3.5 |
| | HM | 56.37 | 54.24 | 58 | 64.85 | 66.8 | 68.16 | **72.20** | +4.0 |
| | Base | 56.48 | 92.19 | 92.5 | 87.49 | 90.2 | 94.07 | **94.50** | +0.5 |
| EuroSAT | Novel | 64.05 | 54.74 | 55.65 | 60.04 | 63.5 | 73.23 | **79.60** | +6.4 |
| | HM | 60.03 | 68.69 | 69.49 | 71.21 | 74.54 | 82.35 | **86.43** | +4 |
| | Base | 70.53 | 84.69 | 84.10 | 82.33 | 82.56 | 83.00 | **87.47** | +4.5 |
| UCF101 | Novel | 77.50 | 56.05 | 57.35 | 73.45 | 75.56 | 78.66 | **79.17** | +0.5 |
| | HM | 73.85 | 67.46 | 68.21 | 77.64 | 78.92 | 80.77 | **83.13** | +2.5 |

**Table 5: Cross-dataset benchmark evaluation. EnPrompt (Ours) achieves overall favorable performance.**

| | Source | Target | | | | | | | | | | |
|---|---|---|---|---|---|---|---|---|---|---|---|---|
| | ImageNet | Caltech101 | OxfordPets | StanfordCars | Flowers102 | Food101 | Aircraft | SUN397 | DTD | EuroSAT | UCF101 | Average |
| Linear probe CLIP | 66.73 | 92.94 | 89.07 | 65.29 | 71.30 | 86.11 | 24.87 | 62.62 | 44.56 | 47.69 | 66.77 | 65.12 |
| CoOp | **71.51** | 93.70 | 89.14 | 64.51 | 68.71 | 85.30 | 18.47 | 64.15 | 41.92 | 46.39 | 66.55 | 63.88 |
| CLIP-Adapter | 71.40 | 93.85 | 89.57 | 64.66 | 68.85 | 85.54 | 18.53 | 64.35 | 41.86 | 46.43 | 66.77 | 64.04 |
| CoCoOp | 71.02 | **94.43** | 90.14 | 65.32 | **71.88** | 86.06 | 22.94 | 67.36 | 45.73 | 45.37 | 68.21 | 65.74 |
| PLOT | 70.15 | 94.60 | 90.23 | 65.41 | 71.97 | 86.32 | 22.87 | 67.22 | 44.99 | 46.57 | 68.32 | 65.85 |
| MaPLe | 70.72 | 93.53 | 90.49 | 65.57 | 72.23 | 86.20 | 24.74 | 67.01 | **46.49** | 48.06 | 68.69 | 66.30 |
| Ours | 71.03 | 93.93 | **91.20** | **65.63** | 71.73 | **86.40** | 25.13 | 67.67 | 46.47 | **48.96** | **69.73** | **66.69** |

**Table 6: Domain generalization. These approaches are trained on imageNet and tested on datasets with domain shifts.**

| | Source | Target | | | | |
|---|---|---|---|---|---|---|
| | ImageNet | -V2 | -S | -A | -R | Avg. |
| Linear probe CLIP | 66.73 | 60.83 | 46.15 | 47.77 | 73.96 | 57.18 |
| CoOp | **71.51** | 64.2 | 47.99 | 49.71 | 75.21 | 59.28 |
| CLIP-Adapter | 71.40 | 64.5 | 47.72 | 49.75 | 75.55 | 59.38 |
| CoCoOp | 71.02 | 64.07 | 48.75 | 50.63 | 76.18 | 59.91 |
| PLOT | 70.15 | 64.17 | 49.15 | 50.83 | 76.5 | 60.16 |
| MaPLe | 70.72 | 64.07 | 49.15 | 50.9 | 76.98 | 60.27 |
| Ours | 71.03 | **64.3** | **49.5** | **51.45** | **77.83** | 60.77 |

**Table 7: Comparison of different templates on Flowers102.**

| Templates | Base Acc | Novel Acc. | HM | GAP |
|---|---|---|---|---|
| 'a drawing of a' | 98.3 | 77.10 | 86.42 | ±0.05 |
| 'a painting of the' | 97.2 | **77.7** | 86.39 | ±0.05 |
| 'a photo of a' | **98.47** | 77 | **86.43** | / |

## 4.2 Training Strategy

In Table 8, we evaluate the performance of different EnLa training strategies as an ablation. In our approach, the EnLa is implemented as a neural network, which is essential to learn all potentially valuable features during the training stage in order to achieve effective generalization [9, 10, 18, 39, 53, 75, 77, 78]. To this end, we focus on training the model for more epochs to learn richer features (row-2), resulting in improved generalization performance.

**Table 8: Comparison of EnLa train strategies. Results are averaged over 11 datasets. HM refers to harmonic mean**

| Training epoch | Base Acc. | Novel Acc. | HM |
|---|---|---|---|
| EnLa (2 epoch) | 83.22 | 75.91 | 79.68 |
| EnLa (30 epoch) | **84.71** | **76.90** | **80.64** |

## 4.3 Visual Embeddings Length

Figure 4 (left) shows the effect of visual embedding length on the harmonic mean. The visual embeddings are learnable, and the text embeddings are non-learning. We observed a significant decrease in HM when the visual embedding length exceeds 4. It suggests that too many learnable visual parameters inherently decrease the ability of V-L alignment. Overall, we have determined that using 4 visual embeddings is the most suitable method.

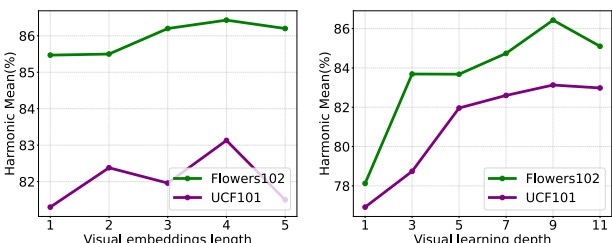

**Figure 4: Ablation study for embedding length and learning depth of image encoder on Flowers102 and UCF101.**

## 4.4 Visual Learning Depth

In Figure 4 (right), we ablate on the visual learning depth for HM. We apply deep prompting on the image encoder. We note that increasing the learning depth generally increases the performance. This suggests that using more layer depth for pre-trained features provides more rich supervision for the features in our approach. The HM value decreases as the number of visual layers increases to 10 or more. It indicates excessive fine-tuning of the model, causing it to lose CLIP generalization ability. Overall, EnPrompt achieves maximum performance at a depth of 9.

## 4.5 Analyzing of Multi-modal Methods

To validate the effect of EnPrompt (Ours), we conducted the HM task and few-shot learning task using two-modal designs and the method completely opposite to EnPrompt in Table 9. We observed that the method (row-3) has a lower performance in novel classes than the method (row-4), which is due to the learnable text embeddings of method (row-3) being weak in generalization compared to the frozen text prompt. Finally, we observed that the method (row-4) and method (row-3) performed better in the novel classes than method (row-2) while still maintaining a few-shot learning task. This is attributed to the fact that method (row-2) is to have neither learnable visual embeddings nor learnable text embeddings. It indicates that prompt learning is crucial for adapting pre-trained VLMs for generalization.

**Table 9: We evaluate EnPrompt with different multi-modal designs. The V-L connection is the default setting.**

| Multi-modal Methods | Base | Novel | HM | Few-shot Task |
|---|---|---|---|---|
| 1: *Visual + Text* | 82.20 | 75.5 | 78.71 | 82.05 |
| 2: Opposite Ours (*Visual*) | 84.15 | 75.20 | 79.43 | 82.69 |
| 3: Ours (*Text*) | **84.71** | **76.90** | **80.64** | **83.27** |

## 4.6 Inference Stage Computational Cost

In Table 10, we show the compute cost analysis of EnPrompt (Ours) and compare it with text embedding learning approaches. Although EnPrompt uses the EnLa, its overall parameters exceed only by 0.52% over CLIP. Compared to MaPLe, EnPrompt has fewer parameters and lower coupling. In terms of inference speed, CoCoOp is significantly slower. In contrast, EnPrompt has no such overhead, obtaining a higher performance with less training time. Although CoOp has a small number of parameters, due to the mismatch of V-L alignment, the training time of 10 epochs for CoOp is similar to ours. EnPrompt is more simpler than text multi-prompt input method (PLOT) in textual design component. Further, EnPrompt provides better convergence as it gets better HM as compared to MaPLe in 10 epochs.

**Table 10: The compute cost comparison using SUN397 dataset. Training time for all methods is calculated for 10 epochs on a single A6000 GPU.**

| Method | Params | Params % CLIP | Train time (min) | HM |
|---|---|---|---|---|
| CoOp | 2048 | 0.002 | 10.88 | 71.65 |
| CoCoOp | 35360 | 0.03 | 39.53 | 75.83 |
| CLIP-Adapter | 0.52M | 0.41 | 8.55 | 72.23 |
| PLOT | 8192 | 0.008 | 10.85 | 76.48 |
| MaPLe | 3.55 M | 2.85 | 10.58 | 79.68 |
| Ours | 0.65M | 0.52 | 10.21 | **80.51** |

## 5 RELATED WORK

## 5.1 Prompt Learning in Vision Language Models

Recently, the strong generalization capability of CLIP [57] has made it a foundation for many methods [22] in adapting pre-trained VLMs for downstream tasks [3, 5, 6, 15, 16, 26, 38, 41, 45, 47, 51, 55, 61, 71, 74]. Prompt learning [37, 40, 42] is a widely used technique in NLP for learning downstream tasks. The use of text prompts, which are instructions provided to the language branch of a Vision-Language model (VLM), is a common practice to enhance task understanding. CoOp [82] and CoCoOp [81] fine-tune the CLIP model specifically for few-shot image recognition by optimizing a continuous set of token embeddings in the language branch. The image-conditional prompt of CoCoOp significantly contributes to enhancing generalization to unseen classes [19, 23, 24] and mitigating the risk of overfitting to the limited labeled data. Moreover, some approaches [44, 72, 73, 83] constrain the proposed learnable prompts to contain the essential general knowledge and prior distribution learning. By conditioning prompts on visual features, CoCoOp [81] ensures that the language model attends to relevant visual information when generating predictions. In addition to single-modal prompt tuning [34], some approaches [35, 43] introduce the multi-modal prompt-tuning designs in CLIP to effectively align V-L representations. However, their methods have not fully released the potential generalization ability of CLIP.

## 5.2 Optimal Transport

Optimal Transport (OT) has recently gained significant attention in various theoretical and application tasks to measure the distance between two probability distributions over metric spaces [1, 2, 29, 63]. For instance, Redko et al.[59] tackle the target shift problem by aligning domain distributions using the OT framework. PLOT [8] introduce a local text feature with textual multi-prompt input. Ours obtain a global text feature for textual one prompt, instead of a local text feature with textual multi-prompt input.

## 6 CONCLUSION

Prompt learning is a promising method for adapting pre-trained vision-language models. However, these methods often struggle to tackle the challenge of generalization on unseen tasks effectively. In this paper, we propose a paradigm called EnPrompt with a novel External Layer (EnLa). Specifically, we propose a textual external layer and learnable visual embeddings for adapting VLMs to downstream tasks. To align the textual and visual features, we propose a novel two-pronged approach. (a) We introduce the optimal transport as the discrepancy metric to align the vision and text modalities. (b) We introduce a novel strengthening feature to enhance the interaction between these two modalities. Extensive experiments clearly demonstrated the effectiveness of our method. In the future, we will extend our effort to more challenging tasks, such as video moment retrieval [69, 70], and investigate the cross-domain generalization ability [68] comprehensively.

## 7 ACKNOWLEDGEMENT

This work was supported in part by National Natural Science Foundation of China (Grant U22A2094 and 62222117), National Key Research and Development Project of China (Grant 2021ZD0110505), and the Zhejiang Provincial Key Research and Development Project (Grant 2023C01043)

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
