# OpenReview forum: "Advancing Prompt Learning through an External Layer"
_acmmm.org/ACMMM/2024/Conference — MM2024 Poster_

### Official Review · Reviewer_7JxX · 2024-05-14

**Rating:** 4
**Confidence:** 2

**Summary:**

The paper proposes to address the generalization capability of VLMs for unseen tasks by an external layer (EnLa) of text branch and learnable visual embedding of the visual branch. Given the frozen text embeddings, the EnLa is implemented by a single 512x512 linear-ReLU-linear layer empirically. These transformed/adapted text embeddings are employed to modify/fuse with the visual embeddings via a [512, 768] network, referred as V-L function. Then the paper introduces optimal transport as the metric to align the distributions of transformed embeddings from the text and visual branches. The proposed method is evaluated on 4 tasks and 11 datasets.

**Strengths:**

The introduction of the optimal transport metric to align the distribution of transformed text and visual embeddings, and this particular way to transform the text and visual features by the EnLa and V-L function appear somewhat novel to me.

The experiments are thorough and convincing, which advance the SOTA on multiple benchmark datasets.

**Limitations:**

The proposed EnLa appears another simple scheme to adapt the frozen text embeddings, which is not a new technical approach, even though with a new name of external layer.

The technical presentation could be more concise and clearer. Please elaborate the input and output feature dimensionalities of the V-L function at line 398 and its implementation. Sec.2.3 could be moved after Sec.2.4, so both of the inputs to the optimal transport function are explained first.

The proposed method is somewhat novel and readily to implement, though not being presented in a straightforward way. The empirical performance is pretty good and deserves to share with the field.

**Suitability:**

2

---

### Official Review · Reviewer_hhe9 · 2024-05-16

**Rating:** 3
**Confidence:** 4

**Summary:**

This paper proposes a new prompt learning method to adapt the visual-language models to downstream tasks. A learnable external layer is introduced to prevent the degradation of generalization ability to unseen classes. Moreover, optimal transport is used as the discrepancy metric to align the vision and text modalities. Extensive experiments are conducted to validate the effectiveness of the proposed method on 4 settings.

**Strengths:**

This paper is well-written and easy to understand. Extensive experiments are conducted to validate the effectiveness of the proposed method on different settings.

**Limitations:**

1) Authors claim that the poor generalization ability of existing prompt learning methods is the learned embeddings are usually invalid in novel classes. However, the introduced external layer also changes the word embeddings. Why changing the word embeddings in this way can generalize to novel classes?
2) In Eq. 10, the optimal transport is performed on the local visual features. MaskCLIP[1] pointed out that the local visual features of the visual encoder in CLIP cannot encode the region information. Therefore, it is important to make some visualizations of the optimal transport plan.

[1] Zhou C, Loy C C, Dai B. Extract free dense labels from clip[C]//European Conference on Computer Vision. Cham: Springer Nature Switzerland, 2022: 696-712.

3) In MAPLE, the model is trained with fewer epochs (e.g., 5). As shown in Table 9, more epochs are needed to train the EnLa. Do the results in different settings need different training epochs or specific tuned hyper-parameters?

**Suitability:**

3

---

### Official Review · Reviewer_Hkgx · 2024-05-21

**Rating:** 4
**Confidence:** 4

**Summary:**

The paper is about advancing prompt learning methods for adapting pre-trained visual-language models (VLMs) to various downstream tasks. The authors address the challenge of poor generalization performance when applying learned text embeddings to unseen tasks. To overcome this, they propose an approach that introduces an External Layer (EnLa) to the text branch and learnable visual embeddings to the visual branch of the VLMs. This approach is designed to maintain the balance of learning capabilities between the two branches while adapting the models to downstream tasks.

**Strengths:**

This paper address the challenge of poor generalization performance when applying learned text embeddings to unseen tasks and proposed an approach that introduces an External Layer (EnLa) to the text branch and learnable visual embeddings to the visual branch of the VLMs.

**Limitations:**

I believe this is a commendable piece of work. However, I do have some concerns:

W1: The method presented in this paper has achieved commendable performance; however, the approach involves stacking an excessive number of submodules. In contrast, similar work such as CoPrompt[1] achieves excellent results with just alignment, suggesting a more streamlined approach could be equally effective.

W2: The authors should provide a more detailed explanation of their motivations. The current explanation is insufficient.

[1] Consistency-guided Prompt Learning for Vision-Language Models.ICLR, 2024.

**Suitability:**

3

---

### Official Review · Reviewer_ttXa · 2024-05-24

**Rating:** 4
**Confidence:** 4

**Summary:**

The authors focus on the performance-degeneration issue of unseen tasks in prompt learning. Then they propose to introduce an External Layer (EnLa) of text branch, learnable visual embeddings of the visual branch and a two-pronged approach. Extensive experiments on 4 types of representative tasks across11 datasets are conducted to verify the effectiveness of the proposed model.

**Strengths:**

1. The article addresses the performance-degeneration issue of unseen tasks through introducing a novel External Layer (EnLa) and learnable visual embeddings, but freezing the text embeddings.
2. The proposed EnLa introduces a two-pronged approach for aligning the visual and textual features of the two modalities.
3. The evaluation performed is complete.
4. Overall good results when compared to the selected baselines.

**Limitations:**

1. From Figure 2, the paper introduces more complex loss function, external layers and additional learnable visual embeddings, while CoOp only learns text embeddings. It is strange that the training time of EnLa is less than that of CoOp in Table 7.
2. The calculation of row-# between Table 3 and Table 10 seems to be inconsistent.
3. On lines 270-272, the authors argue that “the EnLa can generate drastically different textual representations, leading to a mismatch of text and visual branches”. And then to align two modalities, the paper further introduces a two-pronged approach. It is strange that only integrating EnLa (row-2) significantly outperforms baseline methods (row-1) by a large margin in Table 3.
4. As shown in Equation 3, the prompts are the concatenation between the learnable text embeddings and classes. Therefore the learnable embeddings are class-agnostic. Could you explain why previous works has poor generalization performance?

**Suitability:**

3

---

### Meta-Review · Area_Chair_esA6 · 2024-07-02

**Recommendation:** Accept (Poster)
**Confidence:** 4

**Metareview:**

The authors focus on the performance-degeneration issue of unseen tasks in prompt learning. The paper has many strengths, including addressing the performance-degeneration issue of unseen tasks, a complete evaluation, overall good results when compared to the selected baselines, and the proposed EnLa introducing a two-pronged approach. Although some reviewers still have concerns, such as the lack of comparison between this approach and CoPrompt and EnLa, and it being unclear why changing the word embeddings with the proposed external layer can generalize to novel classes, we hope the authors can thoroughly address these issues in the final version. In conclusion, we have decided to accept this paper.